# Cross-cultural validation of two scales to assess mental wellbeing in persons affected by leprosy in Province 1 and 7, Nepal

Janna I. R. Dijkstra[1]*, Marianne van Elteren[1], Nand Lal Banstola[2], Labhi Shakya[3], Himalaya Sigdel[2], Wim H. van Brakel[4]

**1** Department of Ethics, Law and Humanities, VUmc School of Medical Sciences, Amsterdam University Medical Centres (AUMC), Amsterdam, The Netherlands, **2** NLR Nepal, Biratnagar, Nepal, **3** NLR Nepal, Dhangadhi, Nepal, **4** NLR International, Amsterdam, The Netherlands

* j.i.r.dijkstra@amsterdamumc.nl

**Data Availability Statement:** All data can be found in the manuscript and supporting information files.

## Abstract

To assess mental wellbeing among persons affected by leprosy, this study aimed to validate the Warwick-Edinburgh Mental Wellbeing Scale (WEMWBS) and the Patient Health Questionnaire (PHQ-9, depression tool) in Province 1 and 7, Nepal. Using purposive and convenience sampling, cross-cultural equivalences were assessed through semi-structured interviews with persons affected by leprosy (>18 years). Data were transcribed, translated, analysed and discussed with experts before revising the tools. Psychometric properties of the scales were assessed using an interviewer-administered questionnaire with cases affected by leprosy and controls not affected by leprosy (>18 years). Statistical analysis included internal consistency, construct validity, floor and ceiling effects, and interpretability. The qualitative study included 20 respondents of whom eleven were female. The statements in the original tools were rephrased to questions as participants had difficulties understanding the statements. Six additional changes were made to ensure items were understood well. The quantitative study included 90 cases (46% female) and 50 controls (54% female). The WEMWBS and PHQ-9 had adequate psychometric properties. Cronbach's alphas were 0.85 and 0.76, respectively, indicating good internal consistency, 75% of hypotheses for construct validity were confirmed, no floor and ceiling effects were found, and data to help users interpret results are presented. Our study provides evidence that the adapted versions of the WEMWBS and PHQ-9 have good cultural validity to measure mental wellbeing and depression among persons affected by leprosy in Province 1 and 7, Nepal.

## 1. Introduction

Leprosy or Hansen's disease is not a disease from the past but of the present. Despite an effective and free multidrug treatment, the annual incidence of over 200,000 people worldwide has remained unchanged in the past decades [1, 2]. It is a global health problem as infection by *Mycobacterium leprae* carries a risk of permanent damage to peripheral nerves leading to

**Funding:** The authors received no specific funding for this work.

**Competing interests:** The authors have declared that no competing interests exist.

chronic and often progressive disability [3]. This in turn may cause visible secondary impairments in the form of insensitive limbs and muscle paralysis [4], and due to the well-known stigma, the condition may trigger many social and psychological consequences [5–13]. In fact, the mental impact is often more severe than the physical effects [14–18].

Countries most affected in South-East Asia are India, Indonesia, Bangladesh, and Nepal [2]. Persons affected often face a double burden due to neglect of the debilitating and stigmatizing disease as well as neglect of the associated mental burden. The National Leprosy Control Programme of the Nepalese government does not mention mental health services [19], and, in general, mental health services in the State of Nepal have not received the attention or resources they need [20–22].

Issues associated with leprosy that affect mental wellbeing in Nepal include problems in social functioning, such as divorce, high rates of unemployment, displacement from their homes, and problems involving interpersonal relations, leisure, and social activities. Psychological morbidities that have been described include depression, generalized anxiety disorder, drug and alcohol abuse, and even suicide [23–26].

Persons affected by leprosy are discriminated due to bodily disfigurements which are associated with stigmatizing emotions [18]. Studies have shown that the more severe the (visible) physical disability, the higher the risk of mental distress [9, 17, 18, 26, 27]. In addition, felt stigma and internalisation of stigma cause persons affected by leprosy to feel ashamed, possibly isolating themselves from society [8], and postponing to seek treatment. This may aggravate their condition [23, 27, 28]. A negative self-image increases the mental burden as persons with a more pessimistic image of themselves are less able to cope with physical impairments [29].

In Nepal, as "mental illnesses are on the rise" [30], the Nepal Health Research Council acknowledged "the need for a mental health survey" [31]. This study discusses two instruments to contribute to meeting this need, suitable for use with vulnerable groups like persons affected by leprosy. To ensure instruments are well understood, relevant and acceptable, cultural validation is necessary to thoroughly check the meaning and understanding of the items and establish reliability and validity metrics of such instruments. The objective of this study was to test the cultural validity of the Warwick-Edinburgh Mental Wellbeing Scale (WEMWBS) and the Patient Health Questionnaire (PHQ-9, depression tool) for use with persons affected by leprosy in Province 1 and 7, Nepal. The main research question is: "To what extent are the Nepalese versions of the WEMWBS and PHQ-9 culturally equivalent compared to the original tools?"

## WEMWBS and PHQ-9

Many instruments are available to measure aspects of mental health [32]. For this study, scales WEMWBS and PHQ-9 were selected for cultural validation. The tools are generic, frequently used, and can be administered by a non-specialist. Moreover, they have already been validated in several countries and populations, and they would help in evaluating to what extent mental healthcare is needed [33–36].

The WEMWBS focuses on positive attributes of mental health such as optimism, energy, and confidence [37]. It contains 14 items with response scales ranging from 1 (none of the time) to 5 (all of the time) [38]. A score of 40 or below suggests low mental wellbeing [37, 39]. The tool was shown to be valid and reliable among teenagers, students, and adults in the United Kingdom (UK) [40, 41], and has been validated in countries such as Brazil [42], Spain [43], Italy [44], China [45], and France [46].

The PHQ-9 is a nine-item, reliable, valid measure of depression severity [47]. Each of the nine DSM-IV criteria of depression is scored from 0 (not at all) to 3 (nearly every day), giving

a total score ranging from 0 to 27. A score of 10–14 suggest moderate depression severity, 15–19 moderately severe depression, and 20 or greater severe depression [48]. The PHQ-9 has been validated in South-Korea [49], New Zealand [50], USA [51], Nepal [52], and the Netherlands [53]. Because Kohrt et al. (2016) validated the PHQ-9 in Chitwan, Nepal, we were able to use an abbreviated procedure, just to verify that the tool and translation worked well among persons affected by leprosy.

To assess the construct validity of the tools, a third tool was used in this study. The Explanatory Model Interview Catalogue Stigma Scale for Affected Persons (EMIC-AP) is an existing stigma measure that has been validated in a nearby area in India [54–56], as well as in Brazil [57]. Also, the scale has been used before in Eastern [58] and Western Nepal in Nepali language [59].

## 2. Methods

### Study design

This study used a cross-sectional validation design with a mixed methods approach [60, 61] and was conducted from April until June 2018. Qualitative methods involved semi-structured interviews with persons affected by leprosy. Participants were encouraged to talk about the concepts of leprosy and their psychological health, to answer open questions about their life and their condition, and to comment on the statements of the WEMWBS and PHQ-9. Quantitative methods involved interviewer-administered questionnaires with cases and their controls, who were persons not affected by leprosy. The interview consisted of collecting the items of the two instruments.

### Study population

The study population consisted of persons affected by leprosy living in the vicinity of the cities Biratnagar and Dhangadhi, Nepal. Participants were eligible to include when they were above the age of 18; diagnosed with leprosy, under treatment or had completed treatment; had sufficient command of the Nepali language; could respond independently; and were willing to give informed consent. The control group consisted of persons not affected by leprosy who were family members, friends, neighbours, or community members of cases.

This study aimed to include 20 cases for the qualitative part, or until data saturation was reached, and 100 cases plus 50 controls for the quantitative part, both as separately selected samples and with a male:female ratio of 1. The quantitative sample is based on the minimum number of 100 recommended by Terwee et al. [62].

### Study sampling

NLR Nepal and its partner organisations helped in contacting persons affected by leprosy. Selection techniques involved purposive and convenience sampling. This is acceptable for a validation study, since the most important characteristic of the sample is known and an adequate diversity in the trait that is being assessed is aimed for. The qualitative sample was obtained in the Koshi Zonal Hospital in Biratnagar (Province 1) and the Seti Zonal Hospital in Dhangadhi (Province 7). Persons affected by leprosy were included who were receiving treatment as well as who were called and invited to participate and come to the hospital. The quantitative sample was also obtained via these hospitals and their records, but also via records in Community Health Posts in Province 1 and 7.

Persons affected by leprosy with a range of impairment severity according to the Eye Hand Foot (EHF) score were included [63], since impairment severity was expected to correlate with

mental health status. The EHF score sums up the individual scores for severity and visibility of impairment of each Eye, Hand and Foot, with "0" meaning no visible impairment, anaesthesia or vision loss; "1" only anaesthesia or vision mildly affected; and "2" both visible impairment and anaesthesia, or severe visual impairment. The total EHF score ranges from 0 to 12.

Controls were selected via convenience sampling to ensure that the demographic characteristics were roughly similar to the cases. They were selected at the clinic, door-to-door or on the street. This group was interviewed as a reference group for "normal" mental wellbeing and depression.

## Cultural equivalence testing

Based on Stevelink & van Brakel [60], cultural equivalence was assessed using five categories (see definitions in S1 Table). Conceptual equivalence looked at local conceptions of mental health, the appropriateness of an instrument and theoretical arguments. Semantic equivalence entailed the translation guidelines used, details of the translation procedure, the meaning of key words and phrases, and translation problems and difficulties. Based on the World Health Organization translation guidelines, the process of translation and adaptation of instruments included forward translation, pre-testing, an expert panel, pre-testing and back-translation [64]. Item equivalence evaluated relevance and acceptability of items. Operational equivalence looked at the administration format and response scales. And measurement equivalence focused on validity statistics.

*Internal consistency* measures how well all the items in a tool are correlated. It explored the correlation of a given item with the sum score, indicating whether they assessed the same construct. The indicator was Cronbach's α which would be optimal between 0.70 and 0.90 [62]. *Construct validity* was assessed by formulating several hypotheses per scale (see Table 1). The instruments were compared with each other and with the EHF and EMIC-AP score. If 75% or more of hypotheses were confirmed per instrument, the construct validity was supported [62].

**Table 1. Four pre-defined hypotheses per instrument to confirm construct validity and the results of testing these.**

| Hypothesis | Results |
|---|---|
| WEMWBS | |
| 1. The mean score of the WEMWBS will be significantly lower among cases than controls. | *p*-value ≤ 0.05 (independent T-test) |
| 2. The EMIC-AP score will have a negative correlation with the WEMWBS score. | Divergent validity, ρ −0.20–0.40, i.e., the higher the EMIC-AP score, the lower the WEMWBS score. |
| 3. The PHQ-9 score will have a negative correlation with the WEMWBS score. | Divergent validity, ρ −0.20–0.40, i.e., the higher the PHQ-9 score, the lower the WEMWBS score. |
| 4. The EHF score will have a negative correlation with the WEMWBS score. | Divergent validity, ρ −0.20–0.40 |
| PHQ-9 | |
| 1. The mean score of the PHQ-9 will be significantly higher among cases than controls. | *p*-value ≤ 0.05 (independent T-test) |
| 2. The EMIC-AP score will have a positive correlation with the PHQ-9 score. | Convergent validity, ρ 0.40–0.60, i.e., the higher the EMIC-AP score, the higher the PHQ-9 score. |
| 3. The WEMWBS score will have a negative correlation with the PHQ-9 score. | Divergent validity, ρ −0.20–0.40 |
| 4. The EHF score will have a positive correlation with the PHQ-9 score. | Convergent validity, ρ 0.40–0.60 |

Abbreviations: EHF = Eye Hand Foot; EMIC-AP = Explanatory Model Interview Catalogue Stigma Scale for Affected Persons; PHQ = Patient Health Questionnaire; WEMWBS = Warwick Edinburgh Mental Wellbeing Scale.

*Floor and ceiling effects* were present if 15% or more of the subjects either had the lowest or the highest possible score on the WEMWBS or PHQ-9, indicating low sensitivity at the low or high end of the score [62]. For *interpretability*, to help readers interpret the scores, the mean and confidence intervals (CI) of the WEMWBS and PHQ-9 were calculated in three subgroups based on age, gender, and EHF score. The control group was used as a reference.

## Administration of the scales

The scales were interviewer administered by a trained Nepalese interpreter. To minimise inter-observer variation, all interviews were conducted in Nepali by the same interviewer. Interviewees were first asked to answer or fill in a personal information form. The Principal Investigator was present during all qualitative interviews and almost all quantitative interviews.

## Data collection

The semi-structured interviews were audio recorded. After revision of the tools, the inter-viewer-administered questionnaires were used to test the psychometric properties of the scales. An interview took on average 15–20 minutes. The EMIC-AP was incorporated in the case interviews to test construct validity.

## Data analysis

The tools were forward translated by the interviewer/translator and pre-tested on participants. The semi-structured interviews were transcribed and translated by the interviewer/translator and analysed by the principal investigator. Using framework analysis and coding, interpreta-tions of the statements were divided into being within or out of the range of what the statement was intended to mean [65]. If a majority of participants misinterpreted the question, the items were discussed with an expert panel. Items were revised accordingly and again pre-tested. Before accepting the revised instruments for use in the quantitative study, the final versions of the tools were back translated by an independent translator.

The total scores of the WEMWBS and PHQ-9 (and EMIC-AP) for each subject were summed and a mean sum score was used for further analysis. All personal and quantitative data were entered and managed in Epi Info 7, and statistical analyses were carried out using SPSS statistical software, v25. The mean sum scores were compared between groups by means of a Student's T-test, and data of the control group were used as a reference group for "normal" mental wellbeing and depression. Results were considered significant when the *p*-value was ≤0.05.

## Ethical considerations

Ethical approval was given by the Nepal Health Research Council, reference number 2444. Prior to the interviews, respondents were asked to give written informed consent to participate in the study. All study data were handled with discretion. Personal identifying information was removed from the actual data. Authors did not have access to information that could iden-tify individual participants during or after data collection. Participants' anonymity was assured, and they had the option to withdraw from the interview at any time. No incentives were offered other than a bus fare or compensation for lost wages, if appropriate (max. 500 Nepalese rupees).

Additional information regarding ethical, cultural, and scientific considerations specific to inclusivity in global research is included in the Supporting Information.

## 3. Results

### Participant characteristics

Twenty respondents participated in the qualitative study. The study sample consisted of 11 females and 9 males, all mostly from rural areas (n = 13) in Province 1 (n = 16) and with an EHF score of 0 (n = 17). Median age was 33 years (range 18–63) and 60% attained secondary or higher education (e.g., university) as highest level of education.

The quantitative study included 90 persons affected by leprosy and 50 controls (see Tables 2 and 3). Among the people affected 45.6% was female, the median EHF score was 1 (range 0–12), and the median age was 41 (range 18–80). Among the control group 52.9% was female, and the median age was 40 (range 18–71).

### Conceptual equivalence

The main concepts discussed in the semi-structured interviews were optimism, usefulness, feeling relaxed/tense, interest, energy, decision-making, confidence, respect, concentration, appetite, and feeling down, sad and/or hopeless. Most concepts were perceived similarly in both Western and Nepalese culture.

One concept that was discussed at length was the feeling of being relaxed or tense. Feeling relaxed was correlated with feeling cheerful and *"khusi"*, meaning "happy". When participants

**Table 2. Descriptive information of persons affected by leprosy and controls not affected by leprosy in the quantitative part of the cross-cultural validation study.**

| Characteristic | Cases | | Controls | |
|---|---|---|---|---|
| | n | % | n | % |
| Sample | | | | |
| Male | 48 | 54.4 | 23 | 46 |
| Female | 42 | 46.6 | 27 | 54 |
| Residency | | | | |
| Urban | 16 | 17.8 | 12 | 24 |
| Rural | 74 | 82.2 | 38 | 76 |
| Level of education | | | | |
| Illiterate | 26 | 28.9 | 6 | 12 |
| Read and write only | 30 | 33.3 | 11 | 22 |
| Primary education | 10 | 11.1 | 4 | 8 |
| Secondary education | 23 | 25.6 | 17 | 34 |
| Higher education | 1 | 1.1 | 12 | 24 |
| Income (NPR per month) | | | | |
| No income | 31 | 34.3 | 14 | 28 |
| Less than 7000 | 30 | 33.3 | 9 | 18 |
| 7000–10,000 | 16 | 17.8 | 9 | 18 |
| 10,000–20,000 | 11 | 12.2 | 11 | 22 |
| More than 20,000 | 2 | 2.2 | 7 | 14 |
| EHF score | | | | |
| 0 | 32 | 36.4 | - | - |
| 1 | 21 | 23.9 | - | - |
| 2 | 10 | 11.4 | - | - |
| ≥3 | 25 | 28.4 | - | - |

Abbreviations: EHF = Eye Hand Foot; NPR = Nepalese Rupee; SD = Standard Deviation.

**Table 3. Characteristics of persons affected by leprosy (n = 90) and controls not affected by leprosy (n = 50) in the quantitative part of the cross-cultural validation study.**

| Characteristic | Cases | | Controls | | | |
|---|---|---|---|---|---|---|
| | mean | SD | mean | SD | 95% CI | p-value |
| Age | | | | | | |
| Total | 44.6 | 17.4 | 39.3 | 14.1 | - | - |
| Male | 48.5 | 19.0 | 44.4 | 14.1 | - | - |
| Female | 40.1 | 14.2 | 34.9 | 12.7 | - | - |
| WEMWBS | | | | | | |
| Total | 59.6 | 8.2 | 61.7 | 5.2 | -0.42–4.7 | 0.013* |
| Male | 58.5 | 9.6 | 61.6 | 5.7 | -1.2–7.4 | 0.069 |
| Female | 61.0 | 6.0 | 61.9 | 5.0 | -1.9–3.7 | 0.12 |
| PHQ-9 | | | | | | |
| Total | 4.58 | 3.4 | 4.10 | 2.3 | -1.6–0.60 | 0.25 |
| Male | 4.98 | 4.1 | 3.26 | 2.1 | -3.5–0.11 | 0.096 |
| Female | 4.10 | 2.3 | 4.81 | 2.3 | -0.42–1.9 | 0.91 |
| EMIC-AP | | | | | | |
| Total | 9.40 | 5.8 | - | - | - | - |
| Male | 8.55 | 5.7 | - | - | - | - |
| Female | 10.4 | 5.9 | - | - | - | - |

Abbreviations: CI = Confidence Interval; EMIC-AP = Explanatory Model Interview Catalogue Stigma Scale for Affected Persons; SD = Standard Deviation;

PHQ = Patient Health Questionnaire; WEMWBS = Warwick Edinburgh Mental Wellbeing Scale.

* p ≤ 0.05

felt the opposite of *"khusi"*, the word "tension" was often used by respondents. It could be the result of many things, for example having a problem related to work, family, money, the personal or social situation, study, or concentration.

One concept that showed non-equivalence was appetite, as a connection between feeling bad and eating less or overeating appeared to be lacking. This is different in the original, British culture, where this connection seems to be (more prominently) present [66]. As appetite and depressed feelings do not seem to be related the same in both cultures, it is not equally relevant.

Another concept that showed non-equivalence was confidence, as in Nepali language this is translated as "having faith in yourself". A couple respondents mentioned they are confident because they are healthy or take medicine. It seems that having faith in yourself is related to trust and a feeling that it will be okay. However, in the United Kingdom, the country where the scale was designed, being confident refers more to a feeling of self-assurance, and that you appreciate your own abilities and qualities. Thus, it seems "confidence" does not have the same underlying concept across cultures.

It is interesting to note that in Nepali language there is no word for "depression". When Nepalese persons were asked about how they would describe the feeling of being depressed, concepts were mentioned such as having stress, being anxious, feeling tense or being sad. One woman (60) said she was not depressed at all, as *"she didn't have tension"*.

## Semantic equivalence

Four semantic issues were adjusted. These had sufficient evidence of non-equivalence as the meaning of the translated item differed from the original. For example, in the WEMWBS, Question 4 *("Have you been feeling interested in other people?)* was elaborated with examples

". . . such as family, relatives, neighbours and friends". Without this, some female respondents incorrectly interpreted this question to mean "minding other people's business", for instance on the street or with people you do not know well. They would therefore answer "None of the time" or "Rarely". However, when the extra clause was added, their answers would switch to the answer option "Often" or "All of the time". Male respondents would choose these answer options also without the additional clause.

Also, in Question 10 the word "confident" was changed to "feeling sure of yourself", and answer option b "Rarely" was specified by adding "(1–2 times)". Respondents' understanding was better with these adaptations.

In the PHQ-9, in Question 1 "doing things" the Nepalese translation is "work" (*"kaam"*). This word was changed to "any work/activities" (*"karya"*) to broaden the concept and not only address obligatory things (work), but also things of entertainment and joy, such as eating with family, going to the cinema, etcetera.

## Item equivalence

Only two items in the PHQ-9 scale showed item non-equivalence as they did not appear to be equally relevant or acceptable in both cultures. Data did not show missing elements to be included in the scales.

Question 7 *("Have you been having troubles in concentrating on things, such as reading the newspaper or watching television?")* was adjusted. As not all respondents are literate, one expert proposed to add the clause "or listening to music" at the end. With this addition, illiterate people or people who do not read the newspaper or watch television are given a third option.

Question 9 *("Have you been having thoughts that you would be better off dead or of hurting yourself in some way?")* was also adapted. Several participants had difficulty with the word "dead" and mentioned that this question was direct and somewhat inappropriate. After consultation, it was proposed to reverse the sentence *("Have you been having thoughts of hurting yourself in some way or that you would be better off dead?")*. After this modification, the item was less direct and more suitable in Nepalese culture.

## Operational equivalence

The administration format was not suitable in the original form, as respondents had difficulty comprehending the items as statements. The interviewer had to repeat statements, which was time consuming, and participants misunderstood items. When the interviewer rephrased them as questions participants understood, indicating that the statement format was the issue rather than the translation. All items were therefore changed to questions in both scales. Interviewees of the pilot study were happy with this new improved format. See Additional file 2 for the revised instruments in English language, and Additional file 3 in Nepali language.

## Measurement equivalence

**Internal consistency.** Internal consistency of both tools was good (see Table 4). The WEMWBS total score had an alpha of 0.85. When looking at the Item-to-Total correlations per item, it is evident that Question 3 (*"Have you been feeling relaxed?"*) fit poorly with the other items. Removal would increase the overall alpha from 0.85 to 0.88. The PHQ-9 total score had an alpha of 0.76 that fell in the optimal range. There was no indication of poorly fitting items.

**Construct validity.** Construct validity was supported as 75% of hypotheses were confirmed per instrument (see Table 5). Two hypotheses were not supported. Results showed that stigma scores for persons affected by leprosy were not significantly correlated with wellbeing scores:

**Table 4. Item-Total statistics of the WEMWBS and PHQ-9 in Province 1 and 7, Nepal.**

| Quest | WEMWBS | | PHQ-9 | |
|---|---|---|---|---|
| | Corrected Item-Total Correlation | Cronbach's α if Item Deleted | Corrected Item-Total Correlation | Cronbach's α if Item Deleted |
| 1 | 0.58 | 0.84 | 0.50 | 0.73 |
| 2 | 0.70 | 0.83 | 0.35 | 0.76 |
| 3 | −0.012 | 0.88 | 0.46 | 0.74 |
| 4 | 0.45 | 0.84 | 0.55 | 0.73 |
| 5 | 0.73 | 0.82 | 0.28 | 0.77 |
| 6 | 0.71 | 0.83 | 0.57 | 0.73 |
| 7 | 0.65 | 0.83 | 0.45 | 0.75 |
| 8 | 0.50 | 0.84 | 0.32 | 0.76 |
| 9 | 0.22 | 0.85 | 0.55 | 0.73 |
| 10 | 0.55 | 0.84 | - | - |
| 11 | 0.58 | 0.84 | - | - |
| 12 | 0.26 | 0.85 | - | - |
| 13 | 0.67 | 0.83 | - | - |
| 14 | 0.44 | 0.84 | - | - |

Abbreviations: PHQ = Patient Health Questionnaire; WEMWBS = Warwick Edinburgh Mental Wellbeing Scale.

Spearman's $\rho$ was −0.076 ($p$ = 0.48). Results also indicated that the mean score of the PHQ-9 was not significantly higher among cases than among controls.

**Floor and ceiling effects.** No floor or ceiling effects were observed. For the WEMWBS, none of the participants scored the lowest (14) or highest (70) possible score. For the PHQ-9, only three of the 90 respondents scored the lowest (0) possible score (3.3%), and none scored the highest (27) possible score.

**Interpretability.** To help interpret scores, means and 95% confidence intervals are presented for the case group and three subgroups. The control group was used as a reference (see

**Table 5. Four hypotheses per instrument to confirm construct validity and its study outcome.**

| Hypothesis | Study outcome |
|---|---|
| WEMWBS | |
| 1. The mean score of the WEMWBS will be significantly lower among cases than controls. | The case mean of 59.6 was significantly lower than the control mean of 61.7 ($p \le 0.05$). |
| 2. The EMIC-AP score will have a negative correlation with the WEMWBS score. | Spearman's $\rho$ of −0.076 ($p > 0.05$) showed a negative correlation. |
| 3. The PHQ-9 score will have a negative correlation with the WEMWBS score. | Spearman's $\rho$ of −0.37 ($p \le 0.01$) showed a significant negative correlation. |
| 4. The EHF score will have a negative correlation with the WEMWBS score. | Spearman's $\rho$ of −0.39 ($p \le 0.01$) showed a significant negative correlation. |
| PHQ-9 | |
| 1. The mean score of the PHQ-9 will be significantly higher among cases than controls. | The case mean of 4.58 was not significantly higher than the control mean of 4.10 ($p > 0.05$). |
| 2. The EMIC-AP score will have a positive correlation with the PHQ-9 score. | Spearman's $\rho$ of 0.46 ($p \le 0.01$) showed a significant positive correlation. |
| 3. The WEMWBS score will have a negative correlation with the PHQ-9 score. | Spearman's $\rho$ of −0.37 ($p \le 0.01$) showed a significant negative correlation. |
| 4. The EHF score will have a positive correlation with the PHQ-9 score. | Spearman's $\rho$ of 0.36 ($p \le 0.01$) showed a significant positive correlation. |

Abbreviations: EHF = Eye Hand Foot; EMIC-AP = Explanatory Model Interview Catalogue Stigma Scale for Affected Persons; PHQ = Patient Health Questionnaire; WEMWBS = Warwick Edinburgh Mental Wellbeing Scale.

**Table 6. Three subgroups to check interpretability of the WEMWBS and PHQ-9 in Province 1 and 7, Nepal.**

| | Case | | | Control | | |
|---|---|---|---|---|---|---|
| | **n** | **Mean** | **95% CI** | **n** | **Mean** | **95% CI** |
| WEMWBS | | | | | | |
| Total | 90 | 59.6 | 57.9–61.3 | 50 | 61.7 | 60.2–63.2 |
| Gender | | | | | | |
| Male | 49 | 58.5 | 55.7–61.2 | 23 | 61.6 | 59.1–64.0 |
| Female | 41 | 61.0 | 59.1–62.9 | 27 | 61.9 | 59.9–63.9 |
| Age | | | | | | |
| <40 | 49 | 63.7 | 62.7–64.7 | 23 | 63.0 | 61.3–64.6 |
| ≥40 | 40 | 56.4 | 53.6–59.1 | 27 | 60.7 | 58.3–63.2 |
| EHF | | | | | | |
| <3 | 63 | 61.4 | 59.8–63.1 | - | - | - |
| ≥3 | 25 | 54.4 | 50.5–58.4 | - | - | - |
| Score *n (%)* | | | | | | |
| ≤40 | 4 | 4.4 | - | 0 | 0 | - |
| >40 | 86 | 95.6 | - | 50 | 100 | - |
| PHQ-9 | | | | | | |
| Total | 90 | 4.58 | 3.86–5.30 | 50 | 4.10 | 3.44–4.76 |
| Gender | | | | | | |
| Male | 49 | 4.98 | 3.79–6.17 | 23 | 3.26 | 2.35–4.18 |
| Female | 41 | 4.10 | 3.36–4.83 | 27 | 4.81 | 3.92–5.71 |
| Age | | | | | | |
| <40 | 49 | 3.70 | 3.12–4.28 | 23 | 4.30 | 3.52–5.09 |
| ≥40 | 40 | 5.37 | 4.15–6.58 | 27 | 3.93 | 2.87–4.98 |
| EHF | | | | | | |
| <3 | 63 | 4.13 | 3.35–4.90 | - | - | - |
| ≥3 | 25 | 5.92 | 4.25–7.59 | - | - | - |
| Score *n (%)* | | | | | | |
| <10 | 85 | 94.4 | - | 49 | 98 | - |
| 10–14 | 4 | 4.4 | - | 1 | 2 | - |
| 15–19 | 2 | 2.2 | - | 0 | 0 | - |
| ≥20 | 0 | 0 | - | 0 | 0 | - |

Abbreviations: CI = Confidence Interval; EHF = Eye Hand Foot; PHQ = Patient Health Questionnaire; WEMWBS = Warwick Edinburgh Mental Wellbeing Scale.

Table 6). Regarding the WEMWBS score, the mean score for cases was 59.6 and for controls 61.7. The cut-off value indicative of low mental wellbeing is 40 or below [37]. Four out of 90 cases (4.4%) scored 40 or below, whereas zero out of 50 controls (0%).

Regarding the PHQ-9 score, the mean score for cases was 4.58 and for controls 4.20. The international standard suggesting moderate depression is 10–14, which is 15–19 for moderately severe depression. Four cases (4.4%) scored 10–14 and two cases (2.2%) scored 15–19, whereas only one control (2%) scored 10–14.

## 4. Discussion

The purpose of this study was to perform a cross-cultural validation of the WEMWBS and PHQ-9 by assessing their cultural equivalence for persons affected by leprosy in Province 1 and 7, Nepal, compared to the original tools.

## WEMWBS

The instrument and its underlying concepts were found to be appropriate in the target culture. The study supports the notion that (mental) wellbeing is conceptualised in a similar way in British and Nepalese culture. These findings are comparable with other validation studies of WEMWBS conducted in Norway [67], Spain [68], Korea [69], and the Netherlands [33].

Results showed that after two adjustments all items of the tool showed cultural equivalence and subjects understood them correctly. The first revision was Question 4 *("Have you been feeling interested in other people?")*, which was elaborated with examples *(". . . such as family, relatives, neighbours and friends")*. Some female respondents misinterpreted the original item as "minding other people's business". Perhaps this difference had something to do with Nepalese culture and the lower position of women in society [70–72]. The second revision was Question 10 *("Have you been feeling confident?)*, which was changed to "Have you been feeling sure of myself?". This is comparable to a change made in a Spanish validation study [68].

The Nepali version of the wellbeing tool has adequate psychometric properties according to international standards. The internal consistency of the WEMWBS was very good ($\alpha$ = 0.85). Other validation studies conducted in South Africa, Brazil, UK, Spain, Norway, France, Korea, and the Netherlands found similar results [33, 40, 42, 43, 46, 67–69, 73, 74]. Only Question 3 *("Have you been feeling relaxed?")* performed poorly. Removal of this item would further increase internal consistency ($\alpha$ = 0.88). This result is not found in any other validation study [33, 34, 68]. A possible reason is the Nepali term used for relaxed (*"tanaabmukta"*). This is a less common word that may have been misunderstood by people with low literacy. We recommend therefore that the current meaning of this item should be re-examined and that the term used for 'relaxed' should perhaps be changed (e.g., *"araam"*).

All in all, the WEMWBS is a good choice for measuring mental wellbeing, especially in a target group with low education [75]. In addition, the instrument could be added to quality-of-life assessments, for instance, for persons affected by leprosy [76]. The scope of the concept of quality of life is much wider than the scope of the WEMWBS, but it could be argued that mental wellbeing would be a key outcome to measure in mental healthcare, rehabilitation, stigma reduction, and other programmes aiming to improve quality of life [77, 78].

## PHQ-9

The PHQ-9 was found to be suitable for use among Nepalese-speaking people in South Nepal and representative of the construct of depression, even though the term "depression" does not have a Nepali equivalent. Similar findings emerged in studies conducted in Western Kenya [79], Korea [80], Peru [81], and Nigeria [82]. Most concepts in the WEMWBS and PHQ-9 overlap. Participants did not raise any missing elements. However, another study conducted in Nepal found that somatic complaints could be more reflected in the PHQ-9, not just abdominal complaints [52].

Interestingly, the concept of appetite appeared to be perceived differently in Nepalese and British culture. The Item-to-Total correlation of this Item 5 *("Have you been having a poor appetite or have you been overeating?")* was the lowest in the scale (0.28). This may be because people had little to eat anyway and thus eating less was not an option since they had to work also. Perhaps over-eating was not an option either, because they did not have enough money. An alternative explanation is that appetite changes may only be seen in more severe mood disorders, which we may not have encountered. The study by Kohrt et al. (2016) found a similar result as only Item 5 showed a low item-total correlation. Their hypothesis was that there are high rates of parasitic and gastrointestinal infections in low-income countries which may affect appetite changes. Our recommendation would be to re-examine the concept and possibly rephrase the statement.

Two items of the PHQ-9 were adjusted to reach equivalence. First, the phrase "or listening to music" was added to Question 7 *("Have you been having troubles in concentrating on things, such as reading the newspaper or watching television?")* to give a third option to (illiterate) people who do not read the newspaper and who may not watch television. Second, Question 9 *("Have you been having thoughts that you would be better off dead or of hurting yourself in some way?")* was rephrased to make the item more culturally acceptable; the strong Nepalese phrase concerning suicidal thoughts was put at the end of the sentence instead of at the beginning. Possible reluctance to talk about the subject and its questionable acceptability might have influenced our findings.

Regarding semantic equivalence of the PHQ-9, Kohrt et al. (2016) change Item 1 to "[. . .] how much do you feel that you don't enjoy things, can't enjoy yourself, can't be happy, or don't want to work?". We used a shorter question as we did not have time to investigate which option worked best. A future study could further examine which option would be more adequate to use.

The Nepali version of the PHQ-9 has adequate psychometric properties according to international standards. Internal consistency was good ($\alpha$ = 0.76), confirming findings in other validation studies [47, 79–83]. However no significant difference was found between the mean PHQ-9 scores of cases versus controls. One possible explanation could be that selected controls were not representative of the general population, as they were mainly family members or neighbours of persons affected by leprosy, and thus may also have been affected.

The PHQ-9 measures depression, which in turn could affect overall quality of life, but is also a condition that can be treated. Scores of 10–14 are indicative of moderately severe depression and hence that people need mental health care help that may be given in the community, preferably using psychological treatment methods. People with severe depression should be referred for professional mental health care.

## Cultural equivalence and validity

Our results showed that a good overall concept, item, semantic, and measurement equivalence was achieved for both instruments. It is important to note that conceptual differences may change the relevance of some of the items in assessing mental health in another culture. This should therefore be carefully examined. Operational equivalence of the original instruments was poor as participants were able to comprehend the item well only after the statements had been rephrased to questions. This implies that the statement format rather than the translation was the issue indicating non-equivalence in the format of the items. After this change, a good operational validity was realised. Other validation studies with a statement format have reported similar findings to adapt questionnaire items into an interrogative format [84–86]. Results were unaffected by the change of operational method.

Regarding the response scales, some respondents found it confusing that the two tools used two different sets of answer options. A recommendation is to give the response scales a visual addition, for example a flashcard with appropriate symbols, to make them easier to understand. Adding a pictorial scale was reported to be useful in two other studies conducted in Nepal [52, 84].

Overall, looking at the overall results of the five equivalences, good cultural equivalence was achieved for both the WEMWBS and PHQ-9 after relatively small modifications. They can therefore be considered culturally valid in the Nepalese context to measure mental wellbeing and depression among persons affected by leprosy. The validated tools may be used to collect data about mental wellbeing of persons affected by leprosy and their family members, based on which interventions can be developed to improve their mental wellbeing.

## Limitations

One of the main limitations was that participants may have been too shy or embarrassed to answer personal mental wellbeing questions openly during the interviews. Participants occasionally answered questions with "I do not know" or just a smile. Perhaps emotions and wellbeing are sensitive topics to talk about. To mitigate this effect, the Nepali interpreter would emphasize the safe environment and would kindly encourage participants to speak.

Another limitation was that interviews had to be conducted through a translator. The main risk is that the information acquired might not be precise. To mitigate such potential effects, we focussed on key words and phrases during cross-language transfer, and triangulated multiple data sources to ensure that data interpretation was as reliable as possible, raising the credibility of the study.

## Conclusion

The WEMWBS and PHQ-9 have good cultural validity to assess mental wellbeing and depression in a Nepalese setting, as demonstrated in a sample of persons affected by leprosy. Semantic, conceptual, item, and measurement equivalence were adequate. Operational equivalence was good after the item format was changed from statements to questions.

We hope that the instruments will eventually contribute to improved mental wellbeing of persons affected by leprosy and their families in Nepal. This study contributed to the NMD Toolkit validation project, a project aiming to develop a cross-NTD toolkit of instruments. Future research should confirm the utility of the scales with persons affected by lymphatic filariasis and other neglected tropical diseases.

## Supporting information

**S1 Checklist. STROBE statement–checklist of items that should be included in reports of observational studies.**
(DOCX)

**S1 Table. Definitions of categories to assess cultural equivalence.**
(DOCX)

**S2 Table. Adaptation of item statements of WEMWBS and PHQ-9 in Nepalese culture.**
(DOCX)

**S1 Fig. Final WEMWBS and PHQ-9 questionnaires in Nepali language.**
(PDF)

**S1 Data. Participant characteristics of qualitative and quantitative study.**
(XLSX)

**S1 Text. Semi-structured interviews.**
(DOCX)

**S1 File. Inclusivity in global research.**
(DOCX)

## Acknowledgments

Many thanks to those who contributed and have taken part in this research. Thank you colleagues and friends from NLR Nepal who helped realize this study.

## Author Contributions

**Conceptualization:** Janna I. R. Dijkstra, Marianne van Elteren, Wim H. van Brakel.

**Formal analysis:** Janna I. R. Dijkstra.

**Investigation:** Janna I. R. Dijkstra.

**Methodology:** Janna I. R. Dijkstra, Wim H. van Brakel.

**Project administration:** Nand Lal Banstola.

**Resources:** Marianne van Elteren, Nand Lal Banstola, Labhi Shakya, Himalaya Sigdel, Wim H. van Brakel.

**Supervision:** Marianne van Elteren, Nand Lal Banstola, Labhi Shakya, Himalaya Sigdel, Wim H. van Brakel.

**Validation:** Janna I. R. Dijkstra.

**Visualization:** Janna I. R. Dijkstra.

**Writing – original draft:** Janna I. R. Dijkstra.

**Writing – review & editing:** Janna I. R. Dijkstra, Marianne van Elteren, Nand Lal Banstola, Labhi Shakya, Himalaya Sigdel, Wim H. van Brakel.

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
