## [Decision Letter · Decision Letter 0]

2 Oct 2023

PGPH-D-23-01434

Cross-Cultural Validation of Two Scales to Assess Mental Health in Leprosy-Affected Persons in Province 1 and 7, Nepal

Dear Dr. Dijkstra,

Thank you for submitting your manuscript to PLOS Global Public Health. After careful consideration, we feel that it has merit but does not fully meet PLOS Global Public Health’s publication criteria as it currently stands. Therefore, we invite you to submit a revised version of the manuscript that addresses the points raised during the review process.

Thank you very much for this important study on cross-cultural validation of two scales to assess mental health in leprosy affected people in Nepal. Mental well-being of leprosy affected people is very important due to stigma related to leprosy in countries such as Nepal.  Please work on the comments provided by both reviewers. Here are few of my own observations.

I wonder if the title could contain the word mental well-being than mental health as assessing mental well-being of these patients was mentioned in the background and rationale for this studyYou mentioned in the objective of the study 'to test the cultural validity of the Warwick-Edinburgh Mental Wellbeing Scale (WEMWBS) and the Patient Health Questionnaire (PHQ-9, depression tool) for use with persons affected by leprosy'. These scales can also be used to assess mental well-being among people suffering from other diseases such as HIV which also has an element of stigma. Although in the last concluding sentence on future research using these scales was suggested in lymphatic filariasis and  other neglected tropical diseases, I was curious if it would also be applicable to disease such as HIV. This is my personal opinion/recommendation as I have done some research on stigma related to HIV, but not necessarily needs to be addressed in the discussion or conclusion of this study.

We look forward to receiving your revised manuscript.

Kind regards,

Preeti Mahato, Ph.D.

Academic Editor

Journal Requirements:

2. We ask that a manuscript source file is provided at Revision. Please upload your manuscript file as a .doc, .docx, .rtf or .tex.

Additional Editor Comments (if provided):

Reviewers' comments:

Reviewer's Responses to Questions

**Comments to the Author**

1. Does this manuscript meet PLOS Global Public Health’s publication criteria? Is the manuscript technically sound, and do the data support the conclusions? The manuscript must describe methodologically and ethically rigorous research with conclusions that are appropriately drawn based on the data presented.

Reviewer #1: Yes

Reviewer #2: Yes

2. Has the statistical analysis been performed appropriately and rigorously?

Reviewer #1: Yes

Reviewer #2: Yes

3. Have the authors made all data underlying the findings in their manuscript fully available (please refer to the Data Availability Statement at the start of the manuscript PDF file)?

Reviewer #1: Yes

Reviewer #2: Yes

4. Is the manuscript presented in an intelligible fashion and written in standard English?

Reviewer #1: Yes

Reviewer #2: Yes

5. Review Comments to the Author

Reviewer #1: Thanks to the authors for conducting this important research to develop a translated and validated version of the WEMWBS and PHQ-9 tools for use among persons affected by leprosy in the Nepalese population.

The study objectives were clear, methodology was well-articulated with appropriate study design and description of the study population with sufficient sample size. The result section was clearly presented with sufficient tables to answer the study objectives. The data presented in this study supports the conclusion.

However, I have some comments for your consideration.

General comments

Kindly change the phrase “leprosy-affected persons”. May be better to use a non-discriminatory terminology like “persons affected by leprosy” which is applicable to any other condition.

Author’s summary

“…the validated tools can be used to quantify the double stigma of leprosy and mental wellbeing in Nepal,…”. What do you mean by double stigma of mental wellbeing? Kindly rephrase to convey intended meaning or context.

Introduction

1. WEMWBS and PHQ-9: “A score of 40 or below suggests low mental wellbeing (37).”

Kindly recheck/confirm as quoted figure and interpretation seems at variance with citation. For instance, do you refer to a ‘mean score’ of 40? When used categorically, is a single absolute value used as a cut-off or as a range of the mean score with the standard deviation?

Also, may be more appropriate to cite the actual WEMWBS guide recommended for scoring, analysing and interpreting the tool.

2. “The Explanatory Model Interview Catalogue Stigma Scale for Affected Persons (EMIC-AP) is an existing stigma measure that has been validated in…”

Kindly provide a brief explanation here as to why you are introducing the EMIC-AP tool e.g., for the purpose of assessing the construct validity of the tools, etc.

Methods

1. Study population: “The quantitative sample is based on the minimum number of 100 recommended by Terwee et al. (2007).”

Please use proper citation style.

2. Correct tenses to read: “Internal consistency measured how well all the items in the tool was correlated.”

3. Data collection: “The semi-structured interviews were audio recorded. After revision of the tools, structured interviews were used to test the psychometric properties of the scales.”

For clarity, kindly replace “structured interviews”. E.g., may be better to clearly state interviewer-administered questionnaire to distinguish from the qualitative component (semi-structured interviews). Make similar correction across entire document.

4. Data analysis: The first 2 sentences of this section differ from the stated method above using the WHO translation guideline. If the 4-step forward translation, pretesting, etc approach was used, summarize accordingly for the qualitative data.

5. Data analysis: Description of quantitative data analysis to assess measurement equivalence is appropriate. However, last sentence on statistical significance may not be relevant.

6. Ethical considerations: On last sentence for compensation for lost wages; Amount given should be stated, if given.

Results

1. Twenty cases: reads better as Twenty respondents. Take note of importance of sensitivity to language.

2. “The study population shows a range of age (18 to 63; median 33)”: Would read better as: Median age was 33 years (range 18 to 63 years) and _% attained __ as highest level of education (or rephrase as appropriate).

3. Interpretability: “The cut-off value suggesting “good wellbeing” is above 40.” Kindly refer to above comment on cut-off value of WEMWBS interpretation.

4. Interpretability: “These findings suggest a slightly poorer mental wellbeing among cases than controls.”

Within this context, these findings (mean scores + confidence intervals) suffice to ascertain interpretability. I do not agree that it also suggests generalization about the wellbeing within any of the study groups which were purposively and conveniently sampled, not randomized. I suggest this sentence should be removed.

5. Interpretability: “These findings also suggest a slightly poorer mental wellbeing among cases than controls.” Please see comment on similar statement above.

Discussions

1. “All in all, the WEMWBS is a good alternative to the quality of life instruments…”

This is a bit unclear though the citations imply a direct reference/comparison with the WHOQOL-BREF; please clarify. If yes however, this does not speak to the study objective so may not be an appropriate conclusion here, based on this study results. Furthermore, while I agree that both tools share similarities, it is not convincing that they are alternatives, as their use should depend on the intended objective or choice of outcome measure.

2. PHQ-9: “Scores of 10-14 are indicative of moderately severe depression and hence that people need mental health care help that may be given in the community, preferably using psychological treatment methods. People with severe depression should be referred for professional mental health care.”

Not relevant to this study’s objective. Please see similar comment above on generalization.

References

Reference #2: Please check and include a valid link.

Reviewer #2: This study has shown that the researchers already used a strong method, since they used both qualitative and quantitative approach for this cultural adaptation instrument. The number of sample was enough and the statistics method appropriate for to achieve the conclusion. However, the part of discussion should be revised since most of it only repetition it is only of the result. It is suggested not to mention the same this twice in the manuscript. Discussion should explain more about the culture as local context for the instrument.

6. PLOS authors have the option to publish the peer review history of their article (what does this mean?). If published, this will include your full peer review and any attached files.

**Do you want your identity to be public for this peer review?** For information about this choice, including consent withdrawal, please see our Privacy Policy.

Reviewer #1: No

Reviewer #2: No

---

## [Editor Report · Decision Letter 1]

5 Jan 2024

Cross-Cultural Validation of Two Scales to Assess Mental Wellbeing in Leprosy-Affected Persons in Province 1 and 7, Nepal

PGPH-D-23-01434R1

Dear Dr Dijkstra ,

We are pleased to inform you that your manuscript 'Cross-Cultural Validation of Two Scales to Assess Mental Wellbeing in Leprosy-Affected Persons in Province 1 and 7, Nepal' has been provisionally accepted for publication in PLOS Global Public Health.

Best regards,

Preeti Mahato, Ph.D.

Academic Editor